# A Flexible Meta Learning Model for Image Registration

**Frederic Kanter**                    FREDERIC.KANTER@MIC.UNI-LUEBECK.DE
**Jan Lellmann**                       JAN.LELLMANN@MIC.UNI-LUEBECK.DE
*Institute of Mathematics and Image Computing*

## Abstract

We propose a trainable architecture for affine image registration to produce robust starting points for conventional image registration methods. Learning-based methods for image registration often require networks with many parameters and heavily engineered cost functions and thus are complex and computationally expensive. Despite their success in recent years, these methods often lack the accuracy of classical iterative image registration and struggle with large deformations. On the other hand, iterative methods depend on good initial estimates and tuned hyperparameters. We tackle this problem by combining effective shallow networks and classical optimization algorithms using strategies from the field of meta-learning. The architecture presented in this work incorporates only first-order gradient information of the given registration problems, making it highly flexible and particularly well-suited as an initialization step for classical image registration.

**Keywords:** Image Registration, Meta-Learning, Numerical Optimization

## 1. Introduction

Finding spatial correspondences between images is a core task in computer vision and machine learning, and medical image analysis in particular. *Image registration* is the process of finding these correspondences and aligning images for comparison. Given a two-dimensional *reference image* $\mathcal{R} : \mathbb{R}^2 \to \mathbb{R}$ and a *template image* $\mathcal{T} : \mathbb{R}^2 \to \mathbb{R}$, the goal is to find a dense deformation field $\varphi_x : \Omega_\mathcal{R} \to \mathbb{R}^2$, parametrized by some vector $x \in \mathbb{R}^n$, that maps points from the reference to the template image domain such that the deformed template image $\mathcal{T} \circ \varphi_x$ is similar to the reference image. Finding a suitable deformation function is classically formulated as an optimization problem,

$$\min_{x \in \mathbb{R}^n} f(\varphi_x), \quad f(\varphi) := \mathcal{D}(\mathcal{R}, \mathcal{T} \circ \varphi) + \gamma \mathcal{S}(\varphi), \tag{1}$$

with a similarity measure $\mathcal{D}$, a regularizer $\mathcal{S}$, and weight $\gamma > 0$. Conventional registration algorithms for obtaining dense deformation maps are commonly based on iterative optimization methods (Oliveira and Tavares, 2014; Sotiras et al., 2013) and PDE-based models such as Large Displacement Diffeomorphic Metric Mapping (LDDMM) (Beg et al., 2005), making them computationally expensive. Moreover, they typically employ local optimization methods that can get stuck in local minimizers. In this work, we propose an architecture that combines iterative methods with a trainable component in order to construct a class of both robust and accurate image registration methods.

**Related work**  The success of neural networks and deep learning in various machine learning tasks (He et al., 2016; Vaswani et al., 2017) has also motivated their use in image registration. Neural networks have been used to predict the initial momentum of LDDMM (Yang et al., 2017), combining conventional and learning based algorithms. Typically networks in an end-to-end fashion are proposed. The design of the models can differ vastly, but most focus on learning the mapping from on image to another directly. The Voxel-Morph architecture (Balakrishnan et al., 2019) uses a convolutional neural network to map image pairs to an aligned deformation field followed by a spatial transformer. In (Hering et al., 2019) strategies from conventional image registration, such as multi-level scaling of deformation fields, are included to deal with large deformations. While these strategies use well-known conventional methods to improve the network's performance, we focus on using a rough network-based estimate to guide conventional methods. Instead of learning the deformation directly from data, the authors of (Niethammer et al., 2019) propose to use neural networks as a regularization tool in a conventional registration model. However, existing end-to-end models typically struggle to predict large deformations, often have network architectures that are focused on the particular application, and have a large number of parameters (Heinrich, 2019). Our approach is motivated by *meta-learning,* i.e., improving the efficiency of existing optimization methods using training data: Hochreiter et al. (Hochreiter et al., 2001) proved that Recurrent Neural Networks can be used to construct *efficiently trainable* optimization methods that can adapt to the problem class. In 2016, Andrychowicz et al. (Andrychowicz et al., 2016) showed that learned optimizers can outperform optimizers with hand-designed update rules, including Stochastic Gradient Descent, RMSprop, ADAM, and Nesterov's Accelerated Gradient Descent, in terms of the achieved loss. Finn et al. (Finn et al., 2017) showed that meta-learning networks can be adapted to most common computer vision tasks due to their generalization properties. A more general concept is discussed in (Adler and Öktem, 2017), where an updating operator is learned. The operator calculates gradient-like information from the given objective and is embedded in a classical gradient descent optimization, yielding excellent results for image reconstruction from projections of simulated CT images.

A recent related approach is proposed in (Hoopes et al., 2021), where the authors train a model to perform the – usually very expensive – process of selecting good hyperparameters for a given registration network. Mok et al. present another approach to deal with this problem in (Mok and Chung, 2021). Their method is a *conditional registration framework* applicable to any CNN-based registration, where the high-dimensional feature maps of the registration network are "conditioned" on (i.e., depend on) a low-dimensional regularization parameter. Both methods aim to reduce user interaction for neural network-based registration by limiting the number of hyperparameters. Our work is partly motivated by the desire to reduce outliers – which would have to be manually corrected – in conventional methods by supplying them with a robust starting point.

**Contribution**  In this work, we combine the robustness and speed of network-based methods with the accuracy of classical image registration. We propose a meta-learning-based strategy to construct hybrid trainable image registration methods that are rooted in classical approaches, but can be specialized on a specific class of image registration problems in a data-driven way to improve robustness.

Our network architecture combines the iterative update scheme from gradient-based optimization with a *Long Short-Term Memory* (LSTM) layer architecture (Hochreiter and Schmidhuber, 1997). The developed method allows to quickly find a good estimate even for large deformations, which, when used as a starting point, can greatly improve the robustness of classical iterative image registration methods.

## 2. Methods

Conventional solvers make use of an iterative scheme to find an approximate minimizer of Equation (1), whereas neural networks utilize their expressive capabilities to do so in one inference step. We combine these two approaches by introducing an iterative scheme to an LSTM network. A conventional gradient-based solver update step for finding a minimizer of $f$ in (1) takes the form

$$x_{k+1} = x_k - \alpha_k B_k \nabla f(x_k) , \tag{2}$$

where $\alpha_k$ denotes the step length, typically found by a line search procedure, and $B_k$ is an optional preconditioning matrix that is typically derived from the Hessian $\nabla f$ or an estimate thereof. As in meta-learning approaches (Andrychowicz et al., 2016), we retain the iterative nature in principal, but employ a neural network $\Theta$ for the – now nonlinear – update:

$$x_{k+1} = x_k + \theta_k, \quad \theta_k := \Theta(\nabla f(x_k)) . \tag{3}$$

By adapting the weights in $\Theta$, this allows to automatically tune the minimization process to a specific problem class, rather than having to rely on a generic strategy such as Newton- or Quasi-Newton approaches.

We focus on the task of *affine image registration,* where the deformations are of the form $\varphi_x(z) := Az + b$ with $A \in \mathbb{R}^{2 \times 2}$ and $b \in \mathbb{R}^2$. Consequently, the iterates $x_k$ consist of the entries of the unknown parameters $A$ and $b$.

When constructing classical image registration models of the form in Equation (1), the choice of the data term $\mathcal{D}$ and regularizer $\mathcal{S}$ that make up the objective function $f$ is crucial and directly affects the quality of the obtained deformation. However, since in our case $f$ only serves as a *guide* to construct an iterative process of the form Equation (3), which can then be trained using any loss and is not bound to finding a good minimizer of $f$, the choice is less crucial, and we use a simple SSD term $\mathcal{D}(R, T) := \|R - T\|_2^2$. We also use $\mathcal{S} = 0$, as the restriction to affine deformations already provides enough regularization.

By expanding $K$ repetitions of Equation (3), we obtain a stacked network architecture that is at least as powerful as the iterative method with a corresponding number of iterations, benefits from the non-linear gradient information on $f$, and adds the potential of non-linear, trainable steps. Note that in this pure form, the network has no direct access to the input images $\mathcal{R}$ and $\mathcal{T}$; any information about the problem enters exclusively via the gradient of the energy function $f$.

**Model** The main building block $\Theta$ of our network starts with a linear projection layer $L_{\text{in}}$ in order to expand the dimension by a factor of 4 to allow a larger number of trainable parameters than affine transformation parameters. For the main step, we slightly depart from Equation (3) and employ an LSTM cell as proposed in (Hochreiter and Schmidhuber, 1997). The LSTM cell is a successful building block for deep recurrent neural networks,

in particular for time-series analysis, and features a *hidden state* that allows to carry over additional history from previous layers. In our case this makes the update depend not only on the current iterate $x_k$ and gradient $\nabla f(x_k)$, but also on the hidden state $h_k$ that can incorporate previous gradient and function value information. The LSTM output $o_k$ is then passed to another linear layer $L_{\text{out}}$.:

$$(h_{k+1}, o_k) = \text{LSTM}(h_k, L_{\text{in}}(\nabla f(x_k))) \in \mathbb{R}^{d_x} \times \mathbb{R}^{d_h} \tag{4}$$

$$\theta_k = L_{\text{out}}(o_k) \in \mathbb{R}^{d_h} \times \mathbb{R}^{d_x} , \tag{5}$$

$$x_{k+1} = x_k + \theta_k, \tag{6}$$

where $d_x$ is dimension of transformation parameters, and $d_h$ is the size of the LSTM hidden states. Our complete network is of the form

$$\Theta_K = L_{\text{out}}(\text{LSTM}_k( \; ... \; \text{LSTM}_0(h_0, L_{\text{in}}(\nabla f(x_0))))) \tag{7}$$

This mirrors classical successful methods with gradient history such as (L-)BFGS (Nocedal and Wright, 2006). Our network only uses gradient information as inputs, which is derived from the distance function $\mathcal{D}$. The images $\mathcal{R}$ and $\mathcal{T}$ are not fed into the network directly, but used to calculate the new distance after each layer update.

The full network consists of $K = 5$ repetitions of the LSTM update equations (4)–(6) with individual weights for the LSTM cells and shared weights for the linear layers $L_{\text{in}}$ and $L_{\text{out}}$. For training, model weights were initialized to implement an approximate identity mapping.

**Loss** We use a supervised learning approach to train the network based on simulated deformations. As opposed to weakly- or self-supervised approaches (Hering et al., 2019; Mok and Chung, 2021), we specifically do *not* use the energy $f$ as the loss, i.e., the network has the freedom to find the optimal registration in terms of *distance to the ground truth deformation map*. This elevates the method above simply constructing a better solver for minimizing $f$ – which would limit the solution quality to that of iterative methods and require a careful choice of the distance and regularizer used in $f$ – and allows to better pursue the underlying primary goal of finding a good registration. Consequently, the loss directly compares the mean squared error (MSE) of the deformation fields:

$$\mathcal{L}(\varphi_{x_K}) = \frac{1}{n} \sum_{i=1}^{n} \|\varphi_{x_K}(z_i) - \varphi_{\bar{x}}(z_i)\|_2^2 \tag{8}$$

where $\{z_i | i \in \{1, \ldots, n\}\}$ is the image grid, $\bar{x}$ denotes the ground truth deformation parameters, $x_K$ is the output of $K$ applications of $\Theta$ as outlined above.

**Training** We trained our model in PyTorch 1.3.1 using the Adam optimizer (Kingma and Ba, 2014) with a learning rate of 0.005. Layers were trained in a successive fashion and hyperparameters were chosen by random search.

## 3. Experimental Results

All experiments were performed on a 2x6-core Intel Xeon Gold 6128 CPU @ 3.40GHz with 24 logical cores and 3x GeForce RTX 2080 Ti GPUs with 11 GB of memory each.

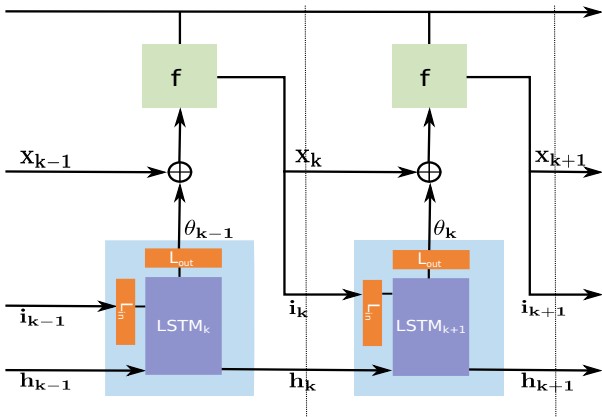

Figure 1: Computational graph for constructing the registration network. The main block $\Theta$ predicts the update $\theta_k$ in Equation (3) and new hidden state $h_k$ from the nonlinear gradient information $i_{k-1} := \nabla f(x_{k-1})$ and the previous hidden state $h_{k-1}$. The update $\theta_k$ is then added to the iterate $x_k$. Repeating this process $K$ times, the resulting stacked network architecture imitates the structure of classical iterative optimization methods, while providing the necessary degrees of freedom to make the process more efficient and robust through training.

**Dataset** For the supervised training approach, we created an artificial data set by deforming a $128 \times 128$ template image, keeping the original image with added noise as reference and the deformed variants as templates. Deformation parameters were chosen randomly from a uniform distribution of rotations in [-60°, 60°] and translations of at most a quarter of the image dimensions. Training samples were created on-the-fly from the given deformation distribution; validation and test data sets were created once and then fixed.

We evaluated the model on three registration tasks: (1) a simple synthetic example of registering x-ray images of two different hands *(Hands)*; (2) a brain data set from the *kaggle* platform originally designed for tumor detection[1], from which registration tasks are created by applying a random transformation to a randomly selected image and adding Gaussian noise *(Brain)*; and (3) DICOM brain images *(fastMRI)* from the NYU fastMRI data set (Zbontar et al., 2018; Knoll et al., 2020), processed in the same way (Figure 2).

**Benchmark Methods** As a simple baseline method *(plain)*, we used a limited-memory BFGS solver with a memory size of $l = 5$ and Armijo line search for minimizing $f$ directly. We limited the maximum number of BFGS iterations to $K = 30$; in our experiments, more iterations did not increase registration quality. The initial estimate for the Hessian was set to the identity matrix scaled by $\frac{1}{\nabla f}$ as proposed in (Nocedal and Wright, 2006). In order to obtain a minimal robustness against local minimizer, the solver is embedded into a coarse-to-fine approach on 4 levels, with image resolutions from $16 \times 16$ to $128 \times 128$.

---

1. https://www.kaggle.com/navoneel/brain-mri-images-for-brain-tumor-detection - visited 2021

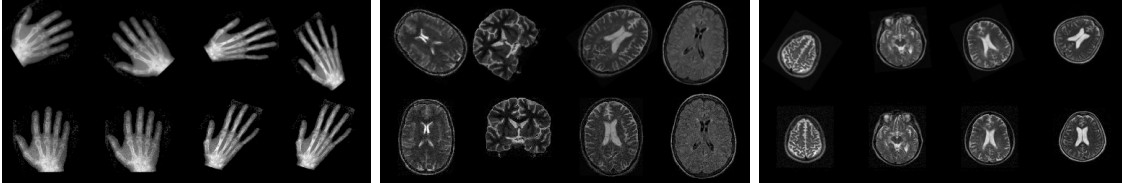

Figure 2: Sample image pairs from *Hands* (left), *Brain* (center), and *fastMRI* (right).

For the combined *(comb)* method, we independently trained and used our trained network in order to compute an initial estimate. This initial estimate was then provided as a starting point to the same algorithm as in *plain*, with the number of iterations reduced to $K = 15$ to further highlight the benefits of improved starting point estimation.

We also compared our our results to the affine image registration implemented in the freely available *FAIR* toolbox (Modersitzki, 2009). Again we chose a multi-scale approach with 4 levels, with no limit on the number of iterations on each level. FAIR uses multiple strategies in order to increase robustness, including an improved estimate of the initial Hessian, filtering and regularized interpolation.

Finally, we compared our method to the *elastix* toolbox (Klein et al., 2010; Shamonin et al., 2014). We performed an extensive grid search on the test set to select the optimal distance metric, optimizer and multi-scale scheme. Reported are the results for the the best-performing L-BFGS solver with the advanced mean squares (MS) as similarity measures, again using a coarse-to-fine approach on 4 levels.

**Performance Metrics** We measured performance primarily in terms of the mean squared error (MSE) between ground truth deformations and predicted deformation field as in Equation (8). Additionally, absolute errors of the obtained parameters $A$ in the Frobenius norm $\|A - A_{gt}\|_F$ and $b$ in the Euclidean norm $\|b - b_{gt}\|_2$ are provided in the appendix.

**Results** As measured by the mean squared errors over the test sets in Table 1, the simple baseline method *(plain)* has poor performance, which can be mostly attributed to outliers caused by suboptimal local minima and failed line search. Similarly, the trained network alone has difficulties achieving an accurate registration. However, with our strategy of using the network's predictions as starting points for the baseline method, the combined performance becomes comparable to the the highly tuned *FAIR* and *elastix* results.

While *FAIR* and *elastix* show good results overall, they fail in a significant number of cases (Figure 3). On the smaller tasks, our combined approach in particular learns to avoid extreme outliers, and entirely eliminates outliers for the *Hands* data set. On the larger *fastMRI* task, performance is less satisfactory, which we attribute to a much higher inter-object variance. We further investigated the influence of a multi-level (coarse-to-fine) network structure, which improves the network performance especially for small data sets (Table 6). Equally, the showed no loss in performance when considering full affine deformations including shearing or scaling (Table 5).

The results shows that our network can predict robust starting points even for simple non-tuned optimizers *(plain)* and large deformations up to 60 degree of rotation and

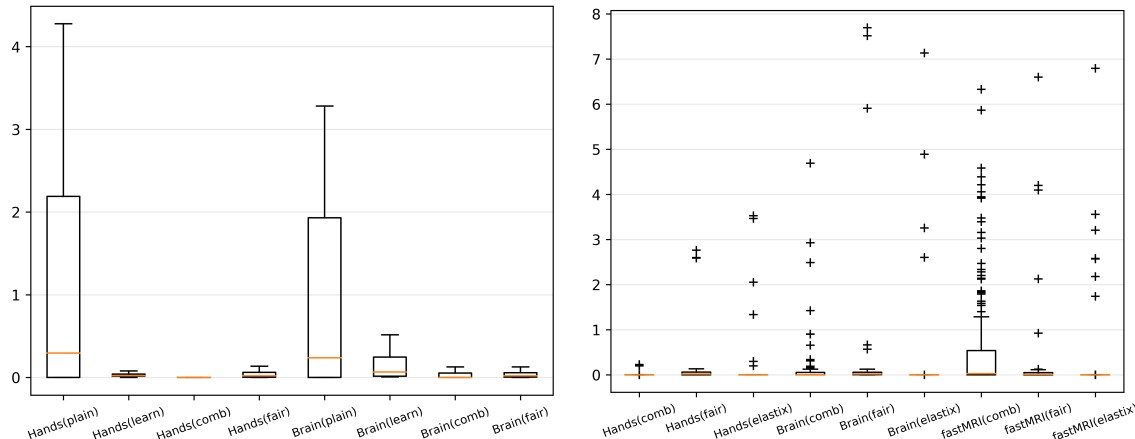

Figure 3: **Left:** When using only either the trained network *(learn)* or the simple baseline method *(plain)* the performance is not competitive. However, when combined *(comb)*, the MSE is comparable to or even better than FAIR with fewer outliers on the *Hands* and *Brain* data sets (see orange median bars; outliers are not shown for clarity). **Right:** MSE of the predicted deformation fields for all tasks. On the smaller *Brain* and in particular *Hands* data sets, our approach allows to reduce the number of extreme outliers and yields a performance comparable to the established methods (see also Table 1). Orange horizontal lines denote medians, boxes show quartiles, and outliers are marked by + symbols.

translations in the range of a quarter of image dimensions. If the distribution of possible deformations can be derived from the data, our method enables good registration results in a few steps without expensive parameter tuning.

Table 1: Mean squared error (MSE) of deformation grids on test sets for *Hands*, *Brain*, and *fastMRI* tasks. By augmenting a simple L-BFGS method *(plain)* with our network *(comb)*, its robustness can be greatly improved, with performance similar to *FAIR* and *elastix* on the specialized tasks (see also Figure 3).

| MSE | Hands | | | | Brain | | | | fastMRI | | | |
|---|---|---|---|---|---|---|---|---|---|---|---|---|
| | plain | comb | FAIR | elastix | plain | comb | FAIR | elastix | plain | comb | FAIR | elastix |
| mean | 1.17 | **0.02** | 0.19 | 0.22 | 1.00 | **0.28** | 0.43 | 0.33 | 0.69 | 0.58 | **0.12** | **0.12** |
| median | 0.30 | **0.00** | 0.02 | **0.00** | 0.24 | **0.00** | 0.02 | **0.00** | 0.07 | 0.03 | 0.01 | **0.00** |
| max | 8.28 | **0.23** | 2.76 | 3.53 | 5.43 | **4.69** | 7.69 | 7.13 | 6.62 | **6.33** | 6.60 | 6.80 |

Table 2: MSE of the predicted deformations for the proposed method *(comb)* when tested on different data sets than used in training. While training on the same class of images as used in the test generates fewer outliers as indicated by the lower mean scores (compare values in Table 1), the network partly generalizes to unseen image classes as indicated by the low median errors. Training on the large *fastMRI* dataset allows to solve the smaller *Hands* and *Brain* problems accurately in many cases.

| training set | Hands | | Brain | | fastMRI | |
|---|---|---|---|---|---|---|
| test set | Brain | fastMRI | Hands | fastMRI | Hands | Brain |
| mean | 1.43 | **0.19** | 0.83 | **0.60** | **0.84** | 1.09 |
| median | 0.22 | **0.00** | 0.11 | **0.05** | 0.16 | **0.01** |
| max | 17.26 | **2.28** | **4.77** | 5.13 | **7.02** | 8.64 |

**Transfer Task** In order to test the generalization of our model to unseen classes of images, we evaluated each model trained on one of the three datasets on the two other data sets that it did not see during training (Table 2). While performance is generally worse than when training on the same data as used for testing (compare Table 1), models trained on the large and most diverse *fastMRI* dataset lead to considerable improvements over the baseline *(plain)*. It will be interesting to see whether it is preferable to train once on a very diverse set, or to re-train for the specific task.

## 4. Conclusion and Outlook

Our image registration strategy combines affine image registration based on iterative conventional registration methods with a robust trainable step for estimating a good initial estimate. By training on the desired class of deformations and image pairs, our method is able to elevate the performance of a simple, accurate, but fragile iterative L-BFGS-based method to a level comparable to methods from the well-established *FAIR* and *elastix* toolboxes that have undergone extensive grid-based parameter search and fine-tuning. In the future, we plan to pursue this promising strategy in particular for non-linear deformations and investigate the effect of more specialized data terms.

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

## Appendix A.

Table 3: Distance $\varepsilon_A := \|A - A_{gt}\|_F$ to ground truth of predicted transformation matrix $A$ for the experiment in Table 1; see also Figure 4–right.

| $\varepsilon_A$ | Hands | | | | Brain | | | | fastMRI | | | |
|---|---|---|---|---|---|---|---|---|---|---|---|---|
| | plain | comb | FAIR | elastix | plain | comb | FAIR | elastix | plain | comb | FAIR | elastix |
| mean | 0.63 | **0.03** | 0.08 | 0.12 | 0.59 | 0.21 | 0.15 | **0.13** | 0.47 | 0.40 | **0.04** | 0.06 |
| median | 0.40 | **0.00** | **0.00** | **0.00** | 0.42 | 0.03 | **0.00** | **0.00** | 0.19 | 0.13 | **0.00** | **0.00** |
| max | 1.68 | **0.33** | 1.38 | 1.57 | 2.01 | **1.83** | 2.38 | 2.30 | 2.08 | **2.17** | 2.21 | 2.21 |

Table 4: Distance $\varepsilon_b := \|b - b_{gt}\|_2$ to ground truth of predicted translation vector $b$ for the experiment in Table 1; see also Figure 4–left.

| $\varepsilon_b$ | Hands | | | | Brain | | | | fastMRI | | | |
|---|---|---|---|---|---|---|---|---|---|---|---|---|
| | plain | comb | FAIR | elastix | plain | comb | FAIR | elastix | plain | comb | FAIR | elastix |
| mean | 0.15 | 0.01 | 0.08 | **0.03** | 0.08 | 0.04 | 0.08 | **0.01** | 0.06 | 0.04 | 0.07 | **0.01** |
| median | 0.13 | **0.00** | 0.07 | **0.00** | 0.03 | 0.01 | 0.07 | **0.00** | 0.03 | 0.02 | 0.06 | **0.00** |
| max | 1.09 | **0.16** | 0.25 | 0.34 | 0.33 | 0.41 | 0.29 | **0.20** | 0.46 | 0.25 | **0.18** | 0.27 |

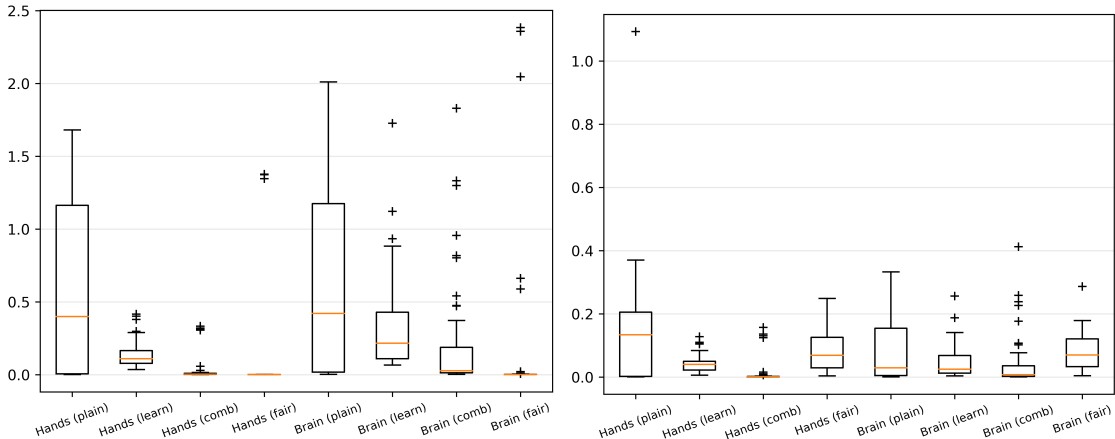

Figure 4: Distance to ground truth of matrix $A$ (left) and translation vector b (right) for the experiment in Figure 3–left; see Table 3 and Table 4.

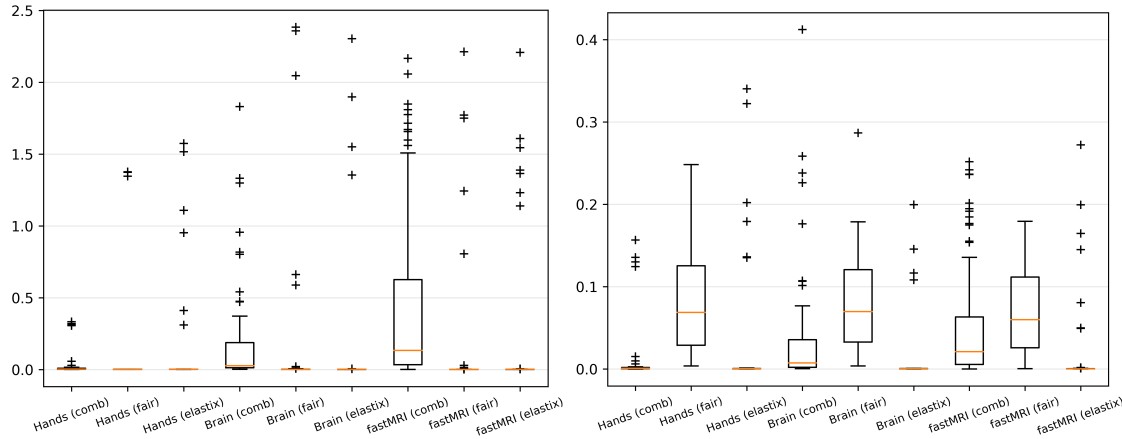

Figure 5: Distance to ground truth of matrix A (left) and translation vector b (right) for the experiment in Figure 3–right.

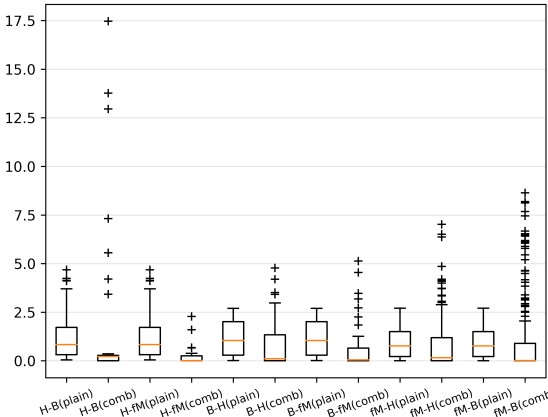

Figure 6: MSE for models tested on different data sets than used for training (see Table 2). Character pairs denote the test and training sets, e.g., *H-B* was trained on *Brain* and tested on *Hands*. The results of the baseline (*plain*) method improve in particularly in terms of the median even for the unseen data sets, at the cost of introducing some additional outliers.

Table 5: Mean squared error (MSE) of deformation grids on test sets for *Hands*, *Brain*, and *fastMRI* tasks. Displayed are the results for full affine transformations: Transformations are created by adding a scalar to each rotation matrix entry randomly picked from a normal distribution (15% of rotation angle). The results confirm the findings in Table 1 for the purely rigid deformations. The *fastMRI* dataset exhibits an extreme outlier, but still the combined (*comb*) method exceeds baseline (*plain*) performance.

| MSE | Hands | | | Brain | | | fastMRI | | |
|---|---|---|---|---|---|---|---|---|---|
| | plain | learn | comb | plain | learn | comb | plain | learn | comb |
| mean | 1.17 | 0.02 | **0.01** | 1.02 | 0.13 | **0.04** | 0.96 | **0.72** | 1.19 |
| median | 0.31 | 0.01 | **0.00** | 0.39 | 0.04 | **0.00** | 0.72 | 0.29 | **0.02** |
| max | 6.55 | 0.13 | **0.23** | 4.51 | 1.54 | **0.56** | **3.02** | 4.97 | 94.30 |

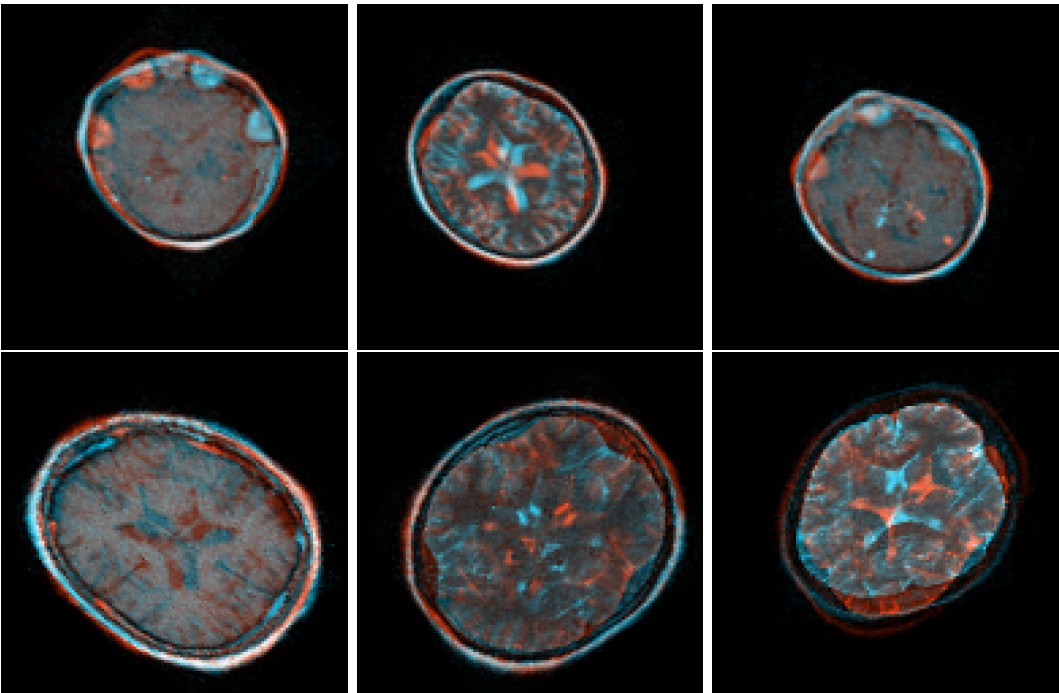

Figure 7: Elastix registration results with the worst MSE for the data sets *fastMRI* (top) and *brain* (bottom). Outliers can be mostly attributed to a high level of symmetry leading to an incorrect initial estimate for the rotation.

Table 6: Mean squared error (MSE) of deformation grids on test sets for *Hands*, *Brain*, and *fastMRI* tasks. Displayed are the results for a multi-level (coarse-to-fine) model architecture. The multi-level structure further improves the results, and for small sets such as *Hands* enables near perfect results.

| MSE | Hands | | | Brain | | | fastMRI | | |
| --- | --- | --- | --- | --- | --- | --- | --- | --- | --- |
| | plain | learn | comb | plain | learn | comb | plain | learn | comb |
| mean | 1.17 | **0.00** | **0.00** | 0.98 | 0.23 | **0.21** | 0.69 | 0.49 | **0.44** |
| median | 0.30 | **0.00** | **0.00** | 0.45 | 0.01 | **0.00** | 0.07 | 0.07 | **0.0** |
| max | 8.28 | 0.03 | **0.00** | **3.19** | 4.18 | 6.43 | **6.62** | 7.85 | 8.97 |

