# OpenReview forum: "A Flexible Meta-Learning Model for Image Registration"
_MIDL.io/2022/Conference — MIDL 2022_

### Official Review · Reviewer_8WAo · 2022-01-14

**Confidence:** 4
**Preliminary Rating:** 2

**Summary:**

The paper proposes a trainable architecture for affine image registration to produce robust starting points for conventional image registration methods. They mainly take advantage of a combination of shallow networks (using LSTM) and classical optimization algorithms (from meta-learning). They looked into the conventional gradient-based solver, and tried to replace the latter term in Eq.(2) with their learnable $\theta_{k}$.

**Strengths:**

The main body of the paper is concise and easy to follow.

They looked into the conventional gradient-based solver, and tried to replace the latter term in Eq.(2) with their learnable $\theta_{k}$. Glad to see an attempt to combine the learning-based approach and the classical methods but also have some concerns (see below).


**Weaknesses:**

My major concerns are the uncertain motivation, confusing arguments and insufficient experiments for their arguments.

I am not sure if this motivation (replacing the previous term as a learnable LSTM-based net) is important and correct for the conventional registration community, and the experiments are performed on simulated data. The proposed method may have a very limited clinical impact. Affine registration is a simple problem and has been extensively studied. To the best of my knowledge, in the operating room, the exact solution of affine image registration is often obtained by using a few matched key points, which are manually allocated by clinicians. This manual point allocation process is efficient and more reliable than (the proposed or other) automatic methods. As a result, the motivation of this work is questionable since it aims to develop a complicated scheme to solve an already solved simple problem.

The argument in the related work section is a little misleading.
For example, we know the regularization weight is crucial, for their mentioned Mok et al. [1], they tried to incorporate the mapping network to search for suitable regularization weight during training rather than time-consuming grid search in non-rigid deformation. While I found this paper’s motivation and application are not in line with their mentioned ones. Only because the paper tried to use an LSTM network to automatically tune the optimization process to a specific problem class? And the paper argued that they reduce human interaction, but, no quantitative measures are utilized to support this. For example, in Mok [1], they measure the time they spent, which indicates the efficiency improvement.
[1] Tony CW Mok and Albert Chung. Conditional deformable image registration with convolutional neural network. MICCAI 2021.

Also, in related work section, authors argue that “the end-to-end (deep learning) models typically struggle to predict large deformations, and are often heavily specialized for the particular application, and have a large number of parameters.” – However, this paper also tried to use an LSTM network to automatically tune the optimization process to a specific problem class (Sec.2). I think it contradicts itself. And, despite with the large number of parameters in deep learning methods, they enable effective learning and fast (even real time) registration during inference. The good efficiency (like VoxelMorph) is why they are popular in research community. Can the proposed method achieve this? I think arguing the large number of parameters is not appropriate in this work, if they are not targeting on a more lightweight deep learning method research. Although they point the limitation of end-to-end deep learning methods, I did not see any end-to-end learning-based affine registration methods are compared. So, I think their argument cannot be sufficiently supported by their experiments.


**Deanonymize Review:**

no

**Detailed Comments:**

Confusing notation: what is your $\mathcal{R}$ for? Most are reference images, but for example, page3-“we also use $\mathcal{R}=0$”, I think here it may be the regularization?

Fig.1 caption: update -> updated

Training process: I am a little confused about the training. Do you combine the classical iterative model and your neural network during training? Or just pretrain your network and embed it into the classical iterative model?


**Final Rating After The Rebuttal:**

3: Borderline

**Justification Of The Final Rating:**

Thanks for the responses from the authors.

I appreciate the reviewers addressed most of my concerns in motivation and methodology, where they rephrased and argued their method is to reduce outliers in affine registration by providing a robust starting point.

Yet, the concerns for the related work description/comparison still could not convince me. For example, they turn to argue the network architecture in the end-to-end model, it is good to see they compare their methods with some end-to-end affine models, even using some simple networks. That is, can this combined method achieve higher performance than the end-to-end affine models with typical architectures? Otherwise, I recommend not arguing these in the paper. Also, if claiming they can provide a better pre-registration for further non-rigid tasks, they have better include such experiment to support it.

But, I appreciate most efforts (also for other reviewers) for the revision, therefore, I change my attitude to neutrality.

**Paper Type:**

methodological development

**Questions To Address In The Rebuttal:**

My major concern is the motivation (see above). Their argument in introduction is a little confusing even misleading for me. I cannot find sufficient experimental results to support their arguments.

Other concerns are expected to clarify.
I am willing to raise my score if the authors can address the above concerns or point the expected information I missed during review.


**Special Issue:**

no

---

### Official Review · Reviewer_iXDt · 2022-01-20

**Confidence:** 3
**Preliminary Rating:** 3
**Recommendation:** Poster

**Summary:**

The paper proposes a new approach that replaces the gradient update in gradient descent optimization with a nonlinear, neural network-based update. The idea is that such an approach can adapt better to a specific optimization problem. The feasibility and performance are demonstrated in an affine image registration task.
The paper is a bit shallow. The idea is nice, but the example in the experiment does not demonstrate its usefulness. I would rather present it in a short paper than a full paper in its current state.

**Strengths:**

The paper addresses an important problem. Optimization of (deep) neural networks is still one of the biggest challenges.  Overall, the paper is well written. It makes you want to see more application examples.

**Weaknesses:**

The “Model” paragraph is a bit difficult to read, and I miss details regarding how the images are fed into the network during training. I mean, I can guess it, but I would like to see if my guess is correct (see detailed comments below). I would also appreciate the publication of the code.
Further, in the experiments, only random rations and translations were used – which covers only some of the degrees of freedom of an affine transform. The abstract, however, suggests that the algorithm can be used for affine image registration.

**Deanonymize Review:**

no

**Detailed Comments:**

First, the term LSTM is used without explanation. And I wonder how close the proposed approach is to an LSTM network. I can see a connection, but there is no further elaboration given in the text.

Figure 1 was a bit difficult to understand. Is it correct that a “main block” contains all three parts Lin+LSTM+Lout? If yes, might be helpful to indicate it in the figure. It might also be helpful to include the loss.

Regarding the results in the experiments: I would like to see some qualitative examples in which elasitx fails. Is it because the structures are too small, and an insufficient amount of point correspondences could be found?

**Final Rating After The Rebuttal:**

4: Weak Accept

**Justification Of The Final Rating:**

I very much appreciate the revision of the manuscript. However, I still think that further experiments are needed. It was not clear to me from the original version of the manuscript that the data was only 2D. However, in the medical field, 3D data is also important. Also, I am not satisfied with the answer regarding "real-world imaging data" (referring to reviewer 8WAo). It may be a common way to evaluate flow models, but in flow models, the subsequent images usually contain the same objects. However, in image registration, we often need to register images from different subjects/with different modalities. Such an experiment would improve the quality of the manuscript.

**Paper Type:**

methodological development

**Questions To Address In The Rebuttal:**

What happens if you augment the dataset for all degrees of freedom of an affine transform?

What is with the image sizes? Many image registration approaches are using a multi-scale technique to determine the (affine) transformation between a pair of images. I wonder, how this aspect was considered in the experiments?

**Special Issue:**

no

---

### Official Review · Reviewer_Ckkt · 2022-01-24

**Confidence:** 3
**Preliminary Rating:** 4
**Recommendation:** Poster

**Summary:**

The authors describe a new method to initialize image registration algorithms. They propose to change the usually linear update step in an optimization procedure with a non-linear step. They learn the non-linear function from the data. The non-linear function is parametrized by a neural network. They conduct experiments on three synthetic image datasets. They compare their method to FAIR toolbox and elastix.

**Strengths:**

The paper is well written and has a clear structure. The mathematical description is very clear. The topic of robust affine registration to initialize non-linear registration algorithms is important for medical image analysis. The experimental setup is interesting and provides a nice starting point for discussions at MIDL.

**Weaknesses:**

I have three main points on motivation, noise model, and outliers.

# Motivation

Here are some points that I think need clarifications. I think this would improve the impact of this paper even outside of the medical image community.

## Introduction

* It's not clear to me why you choose iterative methods over LDDMM. Can you motive that better?
* You write: *However, end-to-end models typically struggle to predict large deformations, are often heavily specialized for the particular application, and have a large number of parameters.*. My comment: Can you back this up with some citations?
* Can you define what you mean by "feasible choice"?
* You write: *can outperform hand-designed methods and are able to generalize well*. My comment: Can you quantify that?
* What is a "conditional registration framework"? What do they condition on?
* What do you mean by "hybrid"?

## Methods

* You write: *stacked network architecture that is at least as powerful as the iterative method*. My comment: Is this obvious? How do you define powerful? Can you quantify that with math or experiments?
* How did you choose $K = 5$? How would you recommend choosing $K$ for a new dataset?
* Is your approach sensitive to the learning rate?

# Noise Model

## Experimental Results — Dataset

* Do you add noise to both the references and deformed images? From what I can tell, you add noise to the template and deform it with a noise-less transformation. Would it makes sense to add noise to the transformation itself or to the transformed image?
* What do you mean by "on-the-fly"?

# Outliers

## Figure 3

* What do these outliers correspond to? Would be interesting to know what went wrong in those cases.

**Deanonymize Review:**

no

**Detailed Comments:**

Some minor points:

* In latex, I would use \text{out} for text
* Strange sentence: *... predicted starting positions. iterations and otherwise did not change any settings compare to the plain baseline. ...*
* Strange sentence: *... baseline method (plain) alone is has poor performance, ...*
* Your write: *Meta-Learning*. My comment: Use same notation throughout the text, e.g. meta learning
* LSTM not defined

**Final Rating After The Rebuttal:**

5: Strong Accept

**Justification Of The Final Rating:**

I thank the authors for replying to all my questions. The authors have adequately addressed my questions. I increased my rating from a weak accept to a strong accept. I congratulate the authors on their work.

**Paper Type:**

methodological development

**Questions To Address In The Rebuttal:**

I would like the authors to address the following three points:

1. Add clarifications (see my comments in *Weaknesses*)
2. Motivate their current noise model or extend it (see my comments in *Weaknesses*)
3. Visualize or report on the outliers (see my comments in *Weaknesses*)

**Special Issue:**

no

---

### Meta-Review · Area_Chair_mfZc · 2022-02-18

**Recommendation:** Accept (Poster)
**Confidence:** 5

**Metareview:**

This paper presents an interesting idea with the novelty of incorporating LSTM into the iterative process of affine registration optimization. Several specific questions were raised by the reviewers, especially related to the motivation of using a recurrent model and details on the method and experiments. The authors’ rebuttal well addressed and clarified those aspects.

---

### Decision · Program_Chairs · 2022-02-28

Accept